# Shared decision making interventions in mental healthcare: a protocol for an umbrella review

Marta Chmielowska [1,2] Yaara Zisman-Ilani [3] Rob Saunders [1] Stephen Pilling [1]

¹Centre for Outcomes Research and Effectiveness (CORE), Research Department of Clinical, Educational and Health Psychology, University College London, London, UK
²Research and Development, The North East London NHS Foundation Trust, London, UK
³Social and Behavioral Sciences, Temple University College of Public Health, Philadelphia, Pennsylvania, USA

**Correspondence to**
Ms Marta Chmielowska;
m.chmielowska@ucl.ac.uk

## ABSTRACT

**Introduction** Shared decision making (SDM) has been advocated as a key component of person-centred care and recovery from mental illness. Although the principles of SDM have been well documented, there is a lack of guidance about how to accomplish SDM in mental healthcare. The objective of the present protocol is to describe the methods for an umbrella review to determine the effectiveness elements of SDM interventions for persons diagnosed with a mental illness. An umbrella review's key characteristic is that it only considers for inclusion the highest level of evidence, namely other systematic reviews and meta-analyses.

**Methods and Analysis** Electronic searches will be performed in CINAHL, PubMed, Scopus, Ovid MEDLINE, Ovid EMBASE, Ovid Cochrane Library, Web of Science, Scopus and Ovid PsycINFO. Based on Joanna Briggs Institute recommended guidelines, review articles will be included if they were published between 2010 and 2021. This approach will help identify current and emerging evidence-based treatment options in mental illness. Included articles will be assessed for quality using Assessment of Multiple Systematic Reviews 2 tool and ratings of the quality of evidence in each review. Presentation of results will align with guidelines in the Cochrane Handbook for Systematic Reviews of Interventions and the Preferred Reporting Items for Systematic Reviews and Meta-Analyses 2020 statement. Findings will be stratified by mode of intervention and implementation characteristics and will inform development of SDM taxonomy in mental healthcare.

**Ethics and dissemination** This umbrella review will focus on the analysis of secondary data and does not require ethics approval. Findings will be disseminated widely to clinicians, researchers and services users via journal publication, conference presentations and social media. The results will contribute to the conceptualisation and understanding of effective SDM interventions in mental healthcare and to improving the quality of SDM for individuals with a mental illness.

**PROSPERO registration number** CRD42020190700.

## Strengths and limitations of this study

► This will be the first umbrella review of systematic review articles about shared decision making (SDM) in mental health.
► This approach will allow for a comprehensive review of a very broad topic by summarising the evidence from multiple research syntheses into one systematic review of reviews.
► The search will be restricted to English and might exclude additional studies published in other languages.
► Assessment of Multiple Systematic Reviews 2 will be used to enable more detailed assessment of systematic reviews that include randomised and/or non-randomised studies of SDM interventions.
► Presentation of results will align with guidelines in the Cochrane Handbook for Systematic Reviews of Interventions and Preferred Reporting Items for Systematic Reviews and Meta-Analyses 2020 statement.

## INTRODUCTION

Shared decision making (SDM) is a health communication approach that focuses on improving patient–clinician interactions around medical decisions in chronic conditions with the ultimate goal of improving clinical and functional outcomes.[1–4] Overall benefits of SDM are well established, including reduced decisional conflict, increased knowledge, satisfaction with care, participation in decision-making, greater treatment engagement and improved clinical outcomes.[5 6] In the last 15 years, SDM has been advocated as the recommended model for treatment and rehabilitation decision making among people affected by mental illness, given that self-determination, choice and autonomy, core principals of an SDM process, are also core aspects of recovery-oriented care.[7–13] Yet, rates of SDM implementation and use in mental health are still very low compared to physical healthcare.[14–16] The literature on SDM in mental illness draws attention to several barriers to SDM implementation, including prevailing stigma among patients and clinicians regarding the patient ability and capacity to make decisions, and issues

related to clinicians' fear of liability and legal exposure.[11 13–16]

Another important barrier to SDM promotion in mental healthcare is the lack of clear definition of SDM practice and the limited understanding of what are the key components of effective SDM interventions in mental health.[10] Currently, and uniquely to mental illness, SDM is interpreted using a wide range of definitions and different types of SDM interventions and practices, which cause confusion and make it hard to standardise SDM as part of a routine mental health practice. Therefore, there is a need to define what is considered an effective SDM approach in mental healthcare and to determine the core elements and steps which are required for its successful implementation in mental health populations. It may be especially the case in situations where the possibility of involuntary hospitalisation creates extreme forms of 'power asymmetry' and where the importance of long-term adherence requires special attention for patient satisfaction with their treatment.[13]

This protocol describes the methods for an upcoming umbrella review to identify and define the effectiveness elements of SDM interventions in mental healthcare, and to support the implementation of SDM principles in clinical practice.

## OBJECTIVES

1. Identify all the recently published systematic reviews and/or meta-analyses which report on the effectiveness of SDM interventions for care and/or treatment of mental health disorders.
2. Assess the scope, and quality of the identified systematic review articles and to provide a more comprehensive account of the available evidence for the effectiveness of SDM, including key components or principles associated with better outcomes.
3. Develop a taxonomic classification of SDM in mental healthcare which will be used as a guide for implementation of evidence-based interventions for care and treatment of mental disorders.

## METHODS

### Protocol and registration

Methods for this umbrella review were developed using criteria for conducting overviews of reviews in the Cochrane Handbook of Systematic Reviews of Interventions. This protocol is registered on the International prospective register of systematic reviews (PROSPERO: CRD42020190700). Only published studies will be examined for this review and no ethical approval is required.

### Patient and public involvement

Patients and/or the public will not be directly involved in this study.

### Eligibility criteria

The reviews considered to be systematic will be included if authors of those reviews defined a strategy to search for studies, to appraise their quality and to synthesise their findings. These may consist of reviews of randomised trials, non-randomised trials and before-and-after studies, including qualitative and observational studies, as long as it helps with understanding the variation in outcomes and the mechanism by which the SDM interventions have an impact. The excluded articles for the current review will consist of non-systematic reviews and studies that involve primary data collection including but are not limited to, randomised trials and non-randomised trials. As an umbrella review, the main focus will be on systematic reviews rather than original studies in order to use the widest range of relevant evidence and compare the best estimates of effectiveness of different interventions. In a situation where the same group of authors published more than one systematic review of the same intervention and patient population, the most recent review will be selected if considered by its authors as an update of their previous review(s). If two or more reviews of the same intervention and patient population are published in a short period of time (<2 years) but with conflicting results, any potential similarities and/or differences will be explored in the full texts of the reviews and lists of included studies. The comparison results will be tabulated, including the rationale for the selection of reviews.

### Quality criteria

To ensure the identified reviews are 'systematic' they would be required to meet the minimum level of methodological rigour, and include studies which addressed the following two items of the Assessment of Multiple Systematic Reviews (AMSTAR) 2 tool[17]: Did the review authors use a comprehensive literature search strategy (eg, were at least two databases searched)? and Did the review authors use a satisfactory technique for assessing the risk of bias (RoB) in individual studies that were included in the review (eg, allocation)? Other umbrella review authors have used similar criteria[18] or limited inclusion to only Cochrane reviews to ensure a minimum level of quality and rigour.[19 20] Therefore, this approach will enhance acceptability and feasibility of the proposed umbrella review.

### Types of interventions

The included reviews may consist of studies where interventions were provided by a wide range of healthcare professionals. Interventions could target patients (eg, patient-mediated interventions), healthcare professionals (eg, distribution of printed educational material) or both (eg, a patient-mediated intervention combined with an intervention targeting healthcare professionals). They could also target patients' families, carers and caregivers (eg, family involvement in care planning) or triads of patients, their family members and healthcare professionals (eg, self-management support). Interventions

could take place in any setting (eg, inpatient, outpatient, primary care, community, secure environment) and will not be restricted by the mode or intensity of delivery. This protocol will rely on the National Institute of Health and Care Excellence (NICE) working definition of SDM which is referred to as a collaborative process through which a healthcare professional supports a person to reach a joint decision about their care.[21] Published journal articles on SDM often do not provide a clear definition, or they use a term inconsistently.[22 23] Thus, the aim of this review is to rectify various definitions (and measurements) and develop a taxonomic classification of SDM in mental healthcare. The included reviews may consist of studies which compared SDM interventions to other interventions with a similar purpose, or with usual care.

### Types of participants
Participants will include adults (aged 18 years and over) who have been diagnosed with a mental health disorder and are facing a decision about their mental health treatment. A mental health disorder will be defined as diagnosable psychological problems which can disrupt thinking, feeling, moods and behaviours, and can cause significant impairment in one's day-to-day functioning. Examples are mood disorders, anxiety disorders, personality disorders, eating disorders, alcohol and substance use disorders, schizophrenia and psychotic disorders. The excluded systematic reviews will target populations other than adults as well as patients diagnosed with mild cognitive impairment, dementia, learning disabilities and an acquired brain injury.

### Outcome measures
The identified SDM practices will be assessed with one of the following types of measures: observer measures, professional-report and/or patient-report tools.[24] A wide range of decision outcomes will be reported and summarised to provide a greater insight into the decision-making process. As proposed by Kreps *et al* in their Transformation Model of Communication and Health Outcomes,[25] patient outcomes will be classified by their impact on the individual across three categories: affective-cognitive, behavioural and physiological. Affective-cognitive outcomes include knowledge, attitudinal and affective/emotional effects. Behavioural outcomes include adherence to recommended treatments and adoption of health behaviours. Physiological outcomes include measures quality of life, self-rated health and biological measures of health.[25] The reviews which extracted all measures of SDM and all mental health outcomes from eligible studies will be included, regardless of the type of outcome measure used or whether the measurement is subjective or objective in nature.

### Search strategy for identification of relevant studies
Nine databases will be searched: CINAHL, PubMed, Scopus, Ovid MEDLINE, Ovid EMBASE, Ovid Cochrane Library, Web of Science, Scopus and Ovid PsycINFO.

References will be managed using Endnote V.X9. The included systematic reviews will be published between 2010 and 2021 and will be limited to English language only. Reviews published before 2010 will be excluded (as per Joanna Briggs Institute guidance), because those published in the past 10 years are considered to represent the contemporary evidence base and will capture primary research conducted over the previous 30 or so years.[26] The search strategy was initially formulated for Ovid Medline (please see online supplemental appendix 1), then further tailored as appropriate for use with other databases. All the drafted and applied search strategies will be publicly available after the review is completed.

### Selection of studies
The initial screening of titles and abstracts will be performed by the primary reviewer (MC) with a random 10% sample screened by a second reviewer (YZ-I). Although a dual-reviewer screening of titles and abstracts is an optimal approach; a single-reviewer screening is an acceptable alternative as stated in the Cochrane Handbook for Systematic Reviews of Interventions.[27] Disagreements will be resolved by discussion between the reviewers, with a senior reviewer acting as arbiter where necessary. Full-text screening of potentially relevant studies will then be performed.

### Data extraction and management
Data will be extracted independently by two reviewers using a previously designed data extraction form. Discrepancies will be resolved by consensus. Where agreement cannot be reached a senior reviewer will consider the paper and a majority decision will be reached. The data extraction form will include the following details where relevant to the study design: an assessment of methodological quality of the included review; the objectives of the review; a summary of the included studies; the interventions studied, the control conditions (if appropriate); the outcomes and time points assessed/evaluated and where relevant estimates of effectiveness, and precision; an assessment of the methodological quality and/or RoB of the included trials and judgements of the quality of the body of evidence. This information will be valuable in order to map the existing evidence. It will also be necessary to identify potential discrepancies in the result of similar reviews.

### Assessment of methodological quality of included reviews
The AMSTAR 2 tool[17] will be used to assess methodological quality of systematic reviews that include both randomised and non-randomised studies of healthcare interventions. The tool provides guidance to rate the overall confidence in the results of a review (high, moderate, low or critically low depending on the number of critical flaws and/or non-critical weaknesses). Given that this is an updated version of AMSTAR, this tool will be preferred for use in future umbrella reviews/overviews. The quality appraisal will include: a table that provides a breakdown

of how each systematic review was rated on each question of the tool, the rationale behind the assessments and an overall rating for each systematic review. The results of the quality/RoB assessments will be then used to help contextualise the umbrella review's evidence base (eg, by assessing whether and to what extent SR methods may have affected the umbrella review's comprehensiveness and results). Two umbrella review authors will assess the quality of each individual text. Discrepancies will be resolved by consensus.

### Assessment of the quality of the evidence in reviews

The Grading of Recommendations Assessment, Development and Evaluation (GRADE) ratings will be extracted from each included review. This approach provides guidance on rating the quality of research evidence in healthcare and has been widely implemented by organisations such as WHO, Cochrane Collaboration, Agency for Healthcare Research and Quality (USA) and NICE (UK). Similar to previous umbrella reviews/overviews, the authors will make judgements to downgrade or upgrade the quality of evidence based on the RoB using criteria specified by the GRADE working group. Discrepancies in the ratings of the quality of evidence will be resolved by consensus between the authors and, if necessary, arbitration by a senior reviewer.

### Data synthesis and presentation

A rigorous international gold-standard methodology of the Preferred Reporting Items for Systematic Reviews and Meta-Analyses (PRISMA) 2020[28] will be employed to facilitate the development and reporting of this protocol. PRISMA 2020 guideline will improve the transparency, accuracy and completeness of the umbrella review protocol. It is expected that the articles will vary considerably both in terms of their review methodology and reporting of outcomes. The presented comparisons will be primarily determined by the content of the included reviews. Data will be grouped where possible according to the population, the type of intervention and outcome measure. Barriers and facilitators for implementation will be identified across different articles and collated. Important limitations within the evidence base will be presented and discussed. Any possible influence of publication/small study biases on review findings will be also considered. Finally, a list of recommendations based on the data synthesis from all studies will be compiled.

### Subgroup analysis

Within the SDM literature, a distinction will be made between interventions targeting patients; interventions targeting patients' families, carers and caregivers; interventions targeting healthcare professionals; interventions targeting patients and healthcare professionals; or interventions targeting clinician-patient-family caregiver triads. This umbrella review will examine the specific nature of SDM interventions used in each context and evaluate if particular types of intervention are more effective for treatment of mental health disorders. The ability to conduct statistical analysis will be dependent on the identified reviews, and how data are presented.

## ETHICS AND DISSEMINATION

Despite widespread support for involving patients with mental illness in decisions about their care, SDM is not yet the norm. Inconsistent definitions and measurement of SDM can complicate efforts to identify the relationships between SDM and patient-reported outcomes and to make any meaningful comparisons across studies. The available evidence for the effectiveness of SDM interventions in mental healthcare is inconclusive when compared with the evidence from the other fields of medicine such as diabetes or cancer.

The present protocol describes the methods and steps for an umbrella review of systematic reviews and meta-analyses of SDM interventions for adults over 18 years of age diagnosed with a mental illness. This review will focus on the analysis of secondary data and is exempt from ethics approval. The target audience consists of clinicians, researchers and service users, who will be reached with tailored materials through journal publications, conference presentations and social media. By facilitating conceptual and practical developments, the review will narrow the current gap between theoretical and policy ideals, and clinical realities in an important area of mental health practice. If the results of this umbrella review are translated into changes in patient care and healthcare practices, then patients will benefit from the reduced burden of hospitalisation either through improved disease treatment and management or better preventative care.

**Contributors** MC wrote the first draft of the protocol. YZ-I and RS read and revised the draft further. SP served as a driving force behind the concept and provided guidance on how to structure the protocol. All authors approved the final version of the manuscript and are accountable for all aspects of the work.

**Funding** This work was supported by the National Institute for Health Research (NIHR), HSDR Project Number: RP-PG-0615-2002.

**Competing interests** None declared.

**Patient and public involvement** Patients and/or the public were not involved in the design, or conduct, or reporting, or dissemination plans of this research.

**Patient consent for publication** Not required.

**Provenance and peer review** Not commissioned; externally peer reviewed.

**ORCID iDs**
Marta Chmielowska http://orcid.org/0000-0003-1987-6187
Yaara Zisman-Ilani http://orcid.org/0000-0001-6852-2583
Rob Saunders http://orcid.org/0000-0002-7077-8729
Stephen Pilling http://orcid.org/0000-0002-7361-8202

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
