## [Reviewer comments · BMJ Open]

ARTICLE DETAILS

TITLE (PROVISIONAL)	Shared decision making interventions in mental health care: A protocol for an umbrella review
AUTHORS	Chmielowska, Marta; Zisman-Ilani, Yaara; Saunders, Rob; Pilling, Stephen

VERSION 1 – REVIEW

REVIEWER	Smith-Merry, Jennifer The University of Sydney, Faculty of Health Sciences
REVIEW RETURNED	02-May-2021

GENERAL COMMENTS	Page 8 Line 122 - why include only reviews containing these study types? This would miss research important to understanding the effectiveness of SDM from the patient perspective via qualitative studies. Page 9 Line 161 - why limit to people over 18? This would make the guidelines that can be developed based on your review more limited, and we need guidelines for children and young people too. Page 9 Line 167 - why exclude these groups if they also have mental illness? Guidelines and new clinical practices also need to work with these groups. Guidelines should be able to be used with groups as broadly as possible. Page 10 Line 196 - best practice would be for two reviewers to screen all titles and abstracts rather than just 10%. Why only 10%? Needs reference here and justification for 10% choice. Given these comments above, the authors need to consider the limitations of their study and include these in the text. Page 8 line 121 - Put 'review' or some other clarifying word before 'authors' in this sentence as I had to reread this line several times because I thought you were referring to yourselves as authors, not the authors of the reviews you are reviewing.
--

REVIEWER	Sequeira, Lydia University of Toronto Institute of Health Policy Management and Evaluation
REVIEW RETURNED	03-May-2021

GENERAL COMMENTS	Thanks for the opportunity to review this paper, it was written clearly and defines the need for a meta-analysis of SDM in mental health due to the lack of standardization of how this is implemented. I am assuming that you have done a preliminary search on this topic and identified several systematic reviews that exist, in order for this kind of an umbrella review to be warranted. A few thoughts and comments for consideration are below:
--

	1. The protocol relies on the NICE working definition of SDM, however, it was also mentioned that there are inconsistent definitions (and measurement) of SDM in the literature - will there be a rectification of the various definitions as one of the results of this umbrella review? How many variations will it include? 2. What search terms (or methodology) will be used in order to ensure that the entire breadth of SDM is included? 3. You mention that comparison results will be tabulated (for similar populations & outcomes) - will there be any statistical analysis conducted on the remainder of the results? 4. In certain SDM protocols across Mental Health care, family member or caregiver involvement is suggested and/or included. From this inclusion criteria, it seems like these studies or reviews may be excluded - do you anticipate a discussion on this piece? Wondering if the the focus on 18 & above is primarily due to this reason (i.e. SDM often involving guardians and/or caregivers)? It could be helpful to add a line for justification. 5. It may be worth collating barriers and facilitators for implementation from all the different review articles, which can help address why there is so much variation across the board. 6. Apart from the exclusion of non-English language studies, are there any other imitations of this study that you anticipate?
--	--

VERSION 1 – AUTHOR RESPONSE

Reviewer: 1

Dr. Jennifer Smith-Merry, University of Sydney

RESPONSE: Thank you for taking the time to review our manuscript and for your valuable comments.

Comments to the Author:

Page 8 Line 122 - why include only reviews containing these study types? This would miss research important to understanding the effectiveness of SDM from the patient perspective via qualitative studies.

RESPONSE: We are grateful for this being highlighted. Our intention is to include systematic reviews of studies involving both RCTs and qualitative research in order to better understand variation in outcomes, the mechanism by which interventions have an impact, and identify ways of tailoring therapy to patient experience. We have clarified this information in the eligibility criteria section: "These may consist of reviews of randomised trials, non-randomised trials, and before-and-after studies (BAs), including qualitative and observational studies, as long as it helps with understanding the variation in outcomes and the mechanism by which the SDM interventions have an impact. The excluded articles for the current review will consist of non-systematic reviews and studies that involve primary data collection including but are not limited to, randomised trials and non-randomised trials. As an umbrella review, the main focus will be on systematic reviews rather than original studies in order to utilise the widest range of relevant evidence and compare the best estimates of effectiveness of different interventions."

Page 9 Line 161 - why limit to people over 18? This would make the guidelines that can be developed based on your review more limited, and we need guidelines for children and young people too.

RESPONSE: The focus of this umbrella review is to summarize accumulative knowledge about adults with mental illness. We fully agree that SDM guidelines for children, youth and family are needed, but it is not part of the aim of the present project. Moreover, parental/guardian consent would require its own specific review and searching protocol. In response to your comment, we have now specified

the focus on adults in the discussion: “The present protocol describes the methods and steps for an umbrella review of systematic reviews and meta-analyses of SDM interventions for adults over 18 years of age diagnosed with a mental illness.”

Page 9 Line 167 - why exclude these groups if they also have mental illness? Guidelines and new clinical practices also need to work with these groups. Guidelines should be able to be used with groups as broadly as possible.

RESPONSE: The focus of this review is to understand how and what in SDM in mental health worked for adults with mental illness. It is true that individuals with dementia may also have comorbid mental illnesses, however, the definition of “mental illness” is broad enough and for the present project, we chose to focus on adults with mental illness and no co-morbidity related to cognitive impairments. We feel that a more specific review protocol would be required to answer questions related to mental health comorbidities with dementia and neurodevelopmental disorders, which is outside of the scope of the current project.

Page 10 Line 196 - best practice would be for two reviewers to screen all titles and abstracts rather than just 10%. Why only 10%? Needs reference here and justification for 10% choice. Given these comments above, the authors need to consider the limitations of their study and include these in the text.

RESPONSE: No consensus exists in method guides regarding whether title and abstract screening should be conducted in duplicate by two reviewers or whether screening by one reviewer is sufficient. Institutions that produce systematic reviews, such as the U.S. Agency for Healthcare Research and Quality (AHRQ)¹, Cochrane², and the National Institute for Health and Care Excellence (NICE)³, acknowledge that dual-reviewer screening is the optimal approach, but view single-reviewer screening as an acceptable alternative. For example, the Cochrane Handbook for Systematic Reviews of Interventions states that “it is acceptable that the initial screening of titles and abstracts is undertaken by only one person.”⁴ Several reviews in SDM in mental health (and other areas) have used the methodology of screening 10% as a team and while reliability is being established, one researcher continues to screen independently.⁵⁻¹⁰ We have clarified this point in the selection of studies section: “The initial screening of titles and abstracts will be performed by the primary reviewer (MC) with a random 10% sample screened by a second reviewer (YZ). Although a dual-reviewer screening of titles and abstracts is an optimal approach a single-reviewer screening is an acceptable alternative as stated in the Cochrane Handbook for Systematic Reviews of Interventions. Disagreements will be resolved by discussion between the reviewers, with a senior reviewer acting as arbiter where necessary. Full text screening of potentially relevant studies will then be performed.” As references for the methods, please see below.

REFERENCES

1. McDonagh M, Peterson K, Raina P, et al. AHRQ Methods for Effective Health Care Avoiding Bias in Selecting Studies. Methods Guide for Effectiveness and Comparative Effectiveness Reviews. Rockville (MD): Agency for Healthcare Research and Quality (US) 2013.
2. Higgins JPT LT, Chandler J, Tovey D, Thomas J, Flemyng E, Churchill R. . Methodological Expectations of Cochrane Intervention Reviews. Cochrane 2019
3. Excellence. NifHaC. Developing NICE guidelines: the manual 2014. . 2014
4. Higgins JPTTJ CJ, Cumpston M, Li T, Page MJ, Welch VA, editors. . Cochrane Handbook for Systematic Reviews of Interventions, Version 6 (updated July 2019); [Chapter 4]: Searching for the selected studies. Cochrane 2019
5. Zisman-Ilani Y, Barnett E, Harik J, et al. Expanding the concept of shared decision making for mental health: Systematic search and scoping review of interventions. Mental Health Review Journal 2017;22(3):191-213. doi: 10.1108/MHRJ-01-2017-0002

6. Barnett ER, Concepcion-Zayas MT, Zisman-Ilani Y, et al. Patient-centered psychiatric care for youth in foster care: a systematic and critical review. *Journal of Public Child Welfare* 2019;13(4):462-89. doi: 10.1080/15548732.2018.1512933
7. Storm M, Husebø AML, Thomas EC, et al. Coordinating Mental Health Services for People with Serious Mental Illness: A Scoping Review of Transitions from Psychiatric Hospital to Community. *Adm Policy Ment Health* 2019;46(3):352-67. doi: 10.1007/s10488-018-00918-7 [published Online First: 2019/01/04]
8. Barnett P, Arundell LL, Saunders R, et al. The efficacy of psychological interventions for the prevention and treatment of mental health disorders in university students: A systematic review and meta-analysis. *J Affect Disord* 2021;280(Pt A):381-406. doi: 10.1016/j.jad.2020.10.060 [published Online First: 2020/11/24]
9. Rosenthal-Oren R, Roe, D., Hasson-Ohayon, I., Roth, S., Thomas, E.C., & Zisman-Ilani, Y. Differences in Causal Beliefs of Psychosis Between People With Psychosis and Mental Health Professionals: A Scoping Review. *Psychiatric Services* In press
10. Thomas EC, Ben-David, S., Treichler, E., Roth S., Dixon, L., Salzer, M., & Zisman-Ilani, Y. . A Systematic Review of Shared Decision Making Interventions for Service Users with Serious Mental Illnesses: State of Science and Future Directions. *Psychiatric Services* In press

Page 8 line 121 - Put 'review' or some other clarifying word before 'authors' in this sentence as I had to reread this line several times because I thought you were referring to yourselves as authors, not the authors of the reviews you are reviewing.

RESPONSE: We added 'of those reviews' after 'authors' to clarify this sentence.

Reviewer: 2

Ms. Lydia Sequeira, University of Toronto Institute of Health Policy Management and Evaluation, Centre for Addiction and Mental Health

Comments to the Author:

Thanks for the opportunity to review this paper, it was written clearly and defines the need for a meta-analysis of SDM in mental health due to the lack of standardization of how this is implemented. I am assuming that you have done a preliminary search on this topic and identified several systematic reviews that exist, in order for this kind of an umbrella review to be warranted.

RESPONSE: Thank you for your feedback and valuable suggestions. Yes, this umbrella review is part of a PhD dissertation by Ms. Chmielowska, and the need for this type of review is based on the gap we identified in the preliminary search and is also based on the SDM in mental health expertise of the co-authors. From initial scoping we know of the existence of some previously conducted systematic reviews of primary studies, and our aim is to identify all existing systematic reviews and develop SDM taxonomy from this evidence base.

A few thoughts and comments for consideration are below:

1. The protocol relies on the NICE working definition of SDM; however, it was also mentioned that there are inconsistent definitions (and measurement) of SDM in the literature - will there be a rectification of the various definitions as one of the results of this umbrella review? How many variations will it include?

RESPONSE: Thank you for the important point. We aim to rectify various definitions (and measurements) and develop a taxonomic classification of SDM in mental healthcare which will be used as a guide for implementation of evidence-based interventions for care and treatment of mental disorders. The number of variations will depend on the results of our search strategy. We have clarified this information in the types of interventions section: "This protocol will rely on the NICE working definition of SDM which is referred to as a collaborative process through which a healthcare professional supports a person to reach a joint decision about their care. Published journal articles on SDM often do not provide a clear definition, or they use a term inconsistently. Thus, the aim of this

review is to rectify various definitions (and measurements) and develop a taxonomic classification of SDM in mental healthcare. The included reviews may consist of studies which compared SDM interventions to other interventions with a similar purpose, or with usual care.”

2. What search terms (or methodology) will be used in order to ensure that the entire breadth of SDM is included?

RESPONSE: We developed separate search strategies for each database. Some of the key terms are decision making, clinical decision making, family decision making, medical decision making, patient decision making, shared decision making, decision support system, clinical decision support system, decision aid, informed choice, informed decision. We addressed this point in the search strategy for identification of relevant studies section, and have included the searches in Appendix 1: “The search strategy was initially formulated for Ovid Medline (please see Appendix 1), then further tailored as appropriate for use with other databases. All the drafted and applied search strategies will be publicly available after the review is completed.”

3. You mention that comparison results will be tabulated (for similar populations & outcomes) - will there be any statistical analysis conducted on the remainder of the results?

RESPONSE: The ability to conduct statistical analysis will be dependent on the identified reviews, and how data is presented. If there is a sufficient number of reviews focused on particular subgroups of individuals or outcomes, then a statistical analysis will be conducted but we have been unable to specify specific analysis in advance. We addressed this point in the subgroup analysis section: “The ability to conduct statistical analysis will be dependent on the identified reviews, and how data is presented.”

4. In certain SDM protocols across Mental Health care, family member or caregiver involvement is suggested and/or included. From this inclusion criteria, it seems like these studies or reviews may be excluded - do you anticipate a discussion on this piece? Wondering if the focus on 18 & above is primarily due to this reason (i.e., SDM often involving guardians and/or caregivers)? It could be helpful to add a line for justification.

RESPONSE: Thank you for the important point. We have now clarified in the types of interventions section and the subgroup analysis section that we will include any SDM interventions (e.g., involving patients, carers, healthcare professionals, and their triads), as long as patients/participants are 18 years and older. Our focus in this project is on adults. Please see also our response to the second comment by Reviewer 1.

5. It may be worth collating barriers and facilitators for implementation from all the different review articles, which can help address why there is so much variation across the board.

RESPONSE: We are very grateful for the reviewer’s suggestion and will collate barriers and facilitators for implementation from all different review articles once the search is complete. We addressed this point in the data synthesis and presentation section: “Barriers and facilitators for implementation will be identified across different articles and collated.”

6. Apart from the exclusion of non-English language studies, are there any other limitations of this study that you anticipate?

RESPONSE: We do not anticipate any significant limitations which can affect our results. The strict exclusion of patients below the age of 18 and those diagnosed with Mild Cognitive Impairment, Dementia, Learning Disabilities, and an Acquired Brain Injury will minimise confounding by such conditions and decrease the inherent population heterogeneity. This approach will result in improved internal validity and will also inevitably limit generalisability of our results.